# Global News Synchrony During the Start of the COVID-19 Pandemic

## ABSTRACT

News coverage profoundly affects how countries and individuals behave in international relations. Yet, we have little empirical evidence of how news coverage varies across countries, languages, locations, political blocs, and time, because of challenges related to measuring and comparing news coverage at a global scale.

To address these challenges, we develop an efficient computational pipeline that comprises three components: 1) a transformer model to estimate multilingual news similarity; 2) a global event identification system that clusters news based on their similarity network; and 3) a method estimating and explaining the synchrony of news across countries and diversity of news within a country, measured based on the news coverage of global events. Each component achieves state-of-the art performance, scaling seamlessly to massive datasets of millions of news articles.

We apply the pipeline to study news articles published between January 1 and June 30, 2020, across 124 countries and 10 languages, and identify the factors explaining biases in national and international news coverage. Our analysis reveals that: (1) news media tend to cover a more diverse set of events in countries that are internally varied: those with federalist governments, larger populations, more official languages, and higher inequality; (2) news coverage is more synchronized between countries that not only actively participate in commercial and political relations—such as, pairs of countries with high bilateral trade volume, and countries that belong to the NATO military alliance or BRICS group of major emerging economies—but also countries that share certain traits—an official language, high GDP, and high democracy indices.

## CCS CONCEPTS

• **Sociology** → **Computational social science**; • **Information systems** → **Web mining**.

## KEYWORDS

global news events, agenda setting, computational social science

## 1 INTRODUCTION

When news media choose to cover a story, they shape the way the public perceives, understands, and discusses current events [48]. By selecting the events that are salient, news media serve a crucial role in shaping the information and political ecosystems. Yet, the

selection of news may differ from location to location even at the same moment, and therefore readers over the globe may be offered different images of the present.

Media and communication scholars offer insights on the factors that may lead news media of different countries to cover similar events. Selection of the news may be driven by the interests of the readers [55], but also by larger forces in the information market such as editorial practices and agenda setting [23, 88]. Across borders, several factors including cultural or political affinity, geographic proximity, and economic relations may impact the synchrony of news coverage [38].

These insights, however, derive mainly from theoretical frameworks and case studies with limited geographic and event coverage. Empirical studies characterized news coverage within a limited set of countries [7, 35], or when they included more countries, they focused on a limited set of news events [2, 87]. Hardly any study compares coverage of events across multiple countries and languages. To obtain insights into media agenda setting at a global scale, there is a need for comprehensive studies of international news coverage.

The key challenges to examining the news coverage at a global scale include the following. First, traditional data collection and validation based on physical newspapers and questionnaires requires human effort that scales linearly with the amount of data and the number of languages. These practical considerations severely limit data size and linguistic coverage of traditional studies. Second, it is not clear how to identify global news events. Most empirical studies focus on few, specific events. Existing methods for identifying which events are covered in the news prioritize precision over coverage, since such methods are based on keywords or entity, or on classification models [11], inevitably leading to the lack of generality. Third, a variety of factors influence the international media market, including economic, political, and linguistic elements. These factors often intertwine, exhibiting complex correlations. Additionally, the measures used to quantify synchrony and diversity of news coverage are not well established. To discern the impact of individual factors on global news coverage, not only it is critical to have a sufficiently large number of samples, it is also crucial to develop robust measures and feature selection techniques.

We overcome these challenges by developing a novel computationally-efficient methodology and applying it to a dataset of 60 million news articles in 10 languages from 124 countries, spanning between January and June 2020. Although the dataset is comprehensive in that it includes global news published in that period, it also includes the onset of the COVID-19 pandemic. This event is uniquely suitable for the study, since it is a rare occurrence of a phenomenon affecting the whole of humanity, and that therefore offers grounds for studying local variations in news coverage.

The main contributions of this work are:

- a computationally-efficient transformer model that infers multilingual news similarity (§4.1),

- a similarity-network-based method for identifying global news events from large news corpora (§4.3),
- a unique information-theoretic approach for measuring country-level synchrony and diversity in news coverage of global events (§5.4), which performs better than a baseline method that does not make use of global events (§5.3),
- by applying the above methods to 2.2 million news articles published across 124 countries, we identify the country-level characteristics that explain international news synchrony and national news diversity globally (§5).

Our results have far-reaching implications for understanding systematic differences in media agenda setting and international relations around the globe. While limited to the first half of 2020 *only*, we offer unique empirical evidence and explanations for news synchrony and diversity across countries. Our methodological contributions enable studying news synchrony and diversity globally across many years and millions of news articles, opening the door for a wave of novel agenda-setting studies. To bolster the effort of academics and media monitoring agencies, we share the data and code of this study.[1]

## 2 RELATED WORK

We contextualize our contributions within a body of related work at the intersection of media studies, computational social science, and machine learning. First, we discuss the societal importance of how news media can set the agenda for the public discourse by deciding what to report on, and how this varies across countries. Then, we delve into the characteristics of countries that may affect the news flow within international media market. Finally, we review existing approaches for quantification of media agenda setting, especially looking at the computational methods that afford comparing news at a global scale.

### 2.1 Agenda setting and newsworthiness

News organizations have to select and prioritize a subset of news to report on each day, a process known as gatekeeping [69]. What the media reports on (and what they do not) has consequences: many studies have shown the ability of the media to "set the agenda" [49, 50]. That is, the events that media select to report on shapes the public dialogue and helps determine what issues are salient. In this way, the media shape "not what to think, but what to think about" [16]. Numerous studies have investigated the values by which news organizations decide what stories are "newsworthy" enough to report on. Beyond the typical effects of how novel, dramatic, or unusual an event is, the spatial and cultural proximity of the event to a media organization's audience is important [28].

### 2.2 International news flow

Beyond studies on which factors shape the choices of individual media organizations, a related body of work investigates how these choices aggregate at a national level to determine patterns of shared media attention and news flow across borders. Grasping how news disseminates across different regions is vital for comprehending

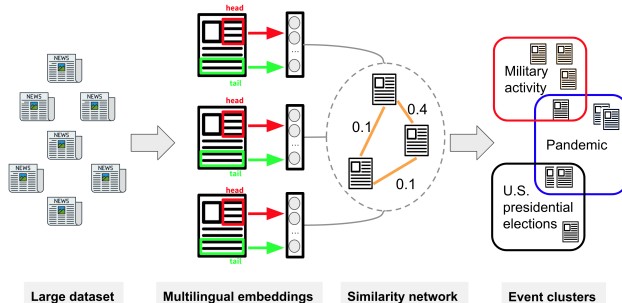

**Figure 1: Our pipeline for global news event detection.**

the global media landscape and its impact on international relations. Mirroring the findings on news newsworthiness, geographic proximity and cultural affinity are found to be important factors, as evidenced by country distance, shared language, religious beliefs, etc. [27, 44, 63, 64, 80, 81]. For example, Semmel [65] studied the coverage of foreign news by four elite US newspapers, and found that those stories that involved a country that was similar to the US economically, politically, or culturally were more likely to be included. Additionally, trade has been identified as a potent factor influencing news connections between nations [30, 44, 63, 81]. Political factors, such as diplomatic relation also play a pivotal role in shaping the news agenda, given that governments can influence journalistic output through regulatory and legislative measures or state-owned media [29, 30, 44]. Due to challenges in analyzing vast amounts of news data, these works remained confined to specific regions. However, computational social science approaches are promising avenues for providing a global perspective about the phenomenon. For example, Zuckerman [88] leveraged the query results from search engine for comparing media attention between countries.

### 2.3 News similarity quantification

Communication scholarship has a longstanding concern in news story diversity. This body of literature argues that news has become less diverse [36, 39], and pointed to two sources of homogenization: journalistic practice—such as the acceleration of the news cycle and increasing reliance on wire service copy—and the monitoring of other media [6]. Qualitative methods like content analysis, used to derive such findings, enabled gaining deep insights while comparing news coverage from multiple outlets [see, e.g., 5, 15, 61]. The difference in how media represent the same news story may be nuanced: it may span several communicative levels from lexical choices to the points of view endorsed by the journalists [68]. However, such qualitative methods are time-consuming and require expertise, thus they are prohibitively costly to perform at a scale, which is necessary for answering questions about news media agenda setting globally. Computational methods are a promising avenue to tackle large-scale analyses [52].

Computational research in this field made several attempts at defining similarity between news articles. These definitions opportunistically fitted a computational task, such as automatically detecting whether two articles cover the same real-world happening [4, 31, 73], story chain [46, 52], event [82, 83], or user preference model [19, 57, 74]. These tasks enabled theoretical and technological

---

[1]Zenodo link hidden to preserve double-blind policy.

advances, such as deduplicating news stories and recommending personalized news feeds. Yet, the task-driven definition of similarity of this body of research limits its applicability to determining the similarity of news articles in a general sense, such as articles that cover stories that are analogous but happen in different locations.

Quantification of news article similarity is a fundamental step that allows comparisons of news articles across countries and over time. Recently, researchers conceptualized and operationalized news similarity as a broader construct that generalizes across event, temporal, and geographic contexts [13]. This broader definition, combined with a large, hand-labeled dataset, enabled NLP researchers to develop machine learning models that achieve human-level performance in estimating multilingual news article similarity.

## 2.4 News event detection

Recent studies also have initiated efforts to detect news events. A common approach involves first obtaining a document representation for each news article [41, 86], e.g. TF-IDF [47], and then clustering the news articles based on similarity metrics of their representations [10, 33, 40]. Leveraging event detection technologies, extensive event data collections have been established to support societal science research, such as ICEWS [75] and GDELT [45]. These collections offer insights of the details and coverage of news events. However, they often limit the event schemes to a *predefined* set, a process that typically demands significant manual effort and is time-consuming in practice. Consequently, these schemes may struggle to adapt to new, emerging events over time. Furthermore, these collections are composed of fine-grain events, e.g., shootings. Thus, they do not capture broader journalistic narratives created around such events.

We take a different approach to news even detection that is not tied to any task or time period, but rather relies on supervised machine learning for news similarity computation and on unsupervised graph clustering for event identification. This approach is computationally efficient and scales well with the number of news articles, enabling a new kind of agenda-setting studies.

## 3 DATASETS

To study international news coverage at scale, we collect a large dataset of news articles published in the first half of 2020. To estimate similarity between news articles from this large dataset, we develop a transformer model by training it on a smaller set of pairs of news articles annotated for their similarity.

## 3.1 Large dataset of news articles

We collect news data from Media Cloud, a platform that maintains a list of news outlets organized by country[2]. This list is diligently updated by communication scholars affiliated with the Media Cloud consortium [59].

To study the phenomenon world-wide, the dataset includes news in ten languages, chosen for their global significance. We adopt the same ten languages as those in the recent work of Chen et al. [13], which also allows us to measure news similarity with high accuracy. To ensure the reliability and validity of our news content from a social science perspective, we excluded all URLs from popular social

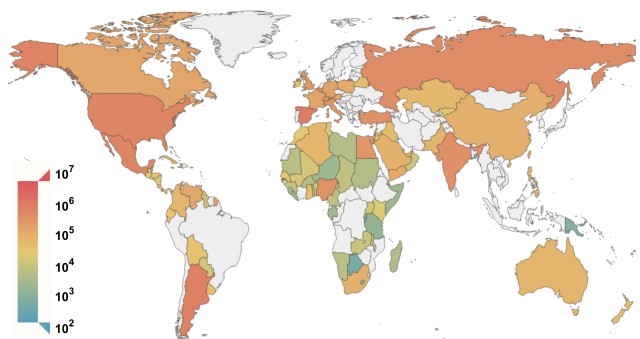

**Figure 2: The number of news articles per country. The color bar is scaled logarithmically.**

media platforms (such as twitter.com, facebook.com, reddit.com, etc). In total, we collected metadata and full text of all news articles from January 1, 2020 to June 30, 2020 in the ten languages, totaling ~60M news articles in the following languages: English (31M articles), Spanish (8.2M), Russian (7.2M), German (3.2M), French (3.2M), Arabic (2.9M), Italian (2.4M), Turkish (1.1M), Polish (595K), and Mandarin Chinese (342K).

## 3.2 Annotated news article pairs

To develop and train a news similarity transformer model, we adopt the dataset and resources provided by a recent study that introduced a nuanced labeling scheme for news similarity [14]. The authors generously shared with us additional data beyond their publicly available dataset from Zenodo.[3] This addition comprises 21,700 (mostly cross-lingual) news article pairs. The extended dataset will be released on Zenodo by the end of 2023.[4]

## 4 METHODOLOGY FOR GLOBAL NEWS EVENT COVERAGE STUDIES

We devise the following novel methodology for global news event coverage (illustrated in Figure 1). First, we develop an efficient transformer model of pariwise news similarity (§4.1). Using this model, we compute similarity between news articles from our large dataset, creating a global news similarity network (§4.2). Then, we cluster this network to identify global news events (§4.3). Once we have such events, we can measure which countries cover each of the events. In this study, we measure whether countries cover a diverse set of events (§5.5) and whether they synchronize in their news event coverage (§5.6).

## 4.1 Multilingual news article similarity model

Measuring news similarity at a global scale is a challenging task, because of the amount of information included in long-form articles, the inherently cross-lingual setting, and the amount of computation it takes. To address these challenges, we refined a multilingual deep learning model to embeds the articles and represents them as numerical vectors. The similarity between news articles is then quantified using the cosine similarity between these vectors. We

---

[2]https://sources.mediacloud.org/#/collections/country-and-state

[3]Dataset: https://zenodo.org/record/6507872#.YtgBhOzMLPY

[4]The final version of this manuscript will include a Zenodo link to the extended dataset.

| Architecture | Model | Performance |
|---|---|---|
| Cross-encoder | **HFL** | 0.811 |
| Bi-encoder | MPNet | 0.492 |
| | MUSE | 0.636 |
| | LaBSE | 0.680 |
| | GateNLP | 0.791 |
| | **This work** | 0.803 |

**Table 1: Performance of pairwise news article similarity models, measured as Pearson correlation on the annotated news dataset, following the prior benchmark [13]. Best models for each architecture are in bold.**

discuss next the model's architecture, and defer implementation details to Appendix A.

In the field of natural language processing, two predominant architectures for text similarity are widely recognized: (1) cross-encoders which ingest two texts as inputs and directly output a similarity score; and (2) bi-encoders, which independently transform each input text into a fixed-length vector, and the similarity is then computed as the closeness between these vectors. We favor the bi-encoder approach because of its computational efficiency: representations for bi-encoders can be pre-computed *once* at an $O(n)$ cost and subsequently used for all comparisons. In other words, the computational cost scales linearly with the number of articles. By contrast, a cross-encoder requires separate computation for each article pair, thus incurring an $O(n^2)$ cost—a computational burden that becomes unsustainable for millions of articles, as in our case.

Recent research suggested that bi-encoders achieve comparable performance to cross-encoders in assessing news similarity [13]. We confirm this observation by comparing the performance of our model on the benchmark developed by Chen et al. [13] against the state-of-art models [13, 17, 20, 70] (Table 1). However, our bi-encoder model is an order of magnitude times faster than cross-encoders, which makes the computation of news similarity across millions of news articles computationally feasible.

## 4.2 Inference on millions of articles

Given that our dataset comprises millions of news articles and, hence, many trillions of news article pairs, despite having an efficient transform model for estimating news similarity, it is still computationally infeasible to compute similarity for every potential pair. However, vast majority of randomly selected news article pairs share little in common. By exploiting this property, we develop a heuristic to identify a smaller set of 13.6 million news article pairs that are more likely to be similar than random pairs and compute news similarity only for this reduced set of pairs. The heuristic preserves the computational viability of this study and it is based on the observation that related articles tend to mention the same named entities, e.g., people, organizations, or locations.

To this end, we extracted named entities from each news article. As those are expressed in natural language, and crucially, in the language of the article, we normalized them by linking them via WikiData, echoing techniques found in other research that utilized Wikipedia for semantic annotation [e.g., 9]. Then, we focused on a subset of articles for which we can compute meaningful named

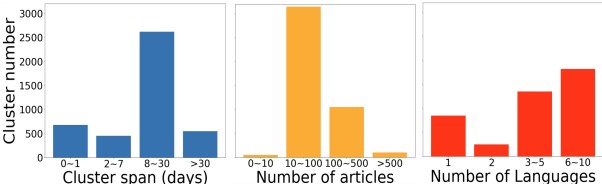

**Figure 3: The histograms of the characteristics of news events: their duration (left), the number of articles (middle), and the numbers of languages (right).**

entity overlap and similarity measures, i.e., those containing a minimum of 100 words (after translating to English) and mentioning at least 10 named entities. This filtering resulted in 15.6 million articles published across 124 countries (visualized in Figure 2). This set of countries covers 64% of all countries and about 76% the world population. To identify pairs of news articles that may be similar, we filter all pairs whose sets of named entities have a Jaccard similarity exceeding 0.25 that were published within a 5-day window of each other. These thresholds are chosen based on the probability that a pair of articles is similar (see details in Appendix B). To illustrate the computational challenge of dealing with such large dataset, we note that the filtering process *alone* took about a week, utilizing thirty 50-core computing nodes equipped with 200 GB RAM each.

Overall, the above steps resulted in a final set of 13.6 million news article pairs across 2.2 million unique news articles. To create a graph of global multilingual news similarity, we applied the transformer model for news similarity on this filtered dataset. The news similarity graph represents articles as nodes and their similarity as weighted edges connecting the nodes.

## 4.3 Identification of global news events

We identify news events by clustering the news similarity network. Thanks to this step, we can move from an *article*-level to an *event*-level analysis of global news. Then, we treat all news articles within a cluster as reporting on the same news event. This level of description has a few advantages. First, news event clusters provide an interpretable summary of the major happening during the studied period of time, as we show in this subsection. Second, the event-level description alleviates the limitations deriving from the heuristics we introduced in the previous subsection to preserve computational feasibility, i.e., article pairs that do not share named entities and, thus, were excluded from news similarity computation can still end up in the same news event cluster. Down the stream, this can lead to more accurate estimates of news diversity and synchrony, which we explain in the following section (§5).

### 4.3.1 Clustering method.
To identify news events, we apply the OSLOM algorithm [42], which identifies statistically significant, overlapping clusters by optimizing a local fitness function that compares the clusters against random fluctuations in a null network model using order statistics. The method has been successfully used in computational social science [26].

### 4.3.2 Results.
Overall, we identified 4357 news events. The majority of events lasted between 1 to 4 weeks and were covered by 10 to 100 news articles (Figure 3). Around 25% of events were covered

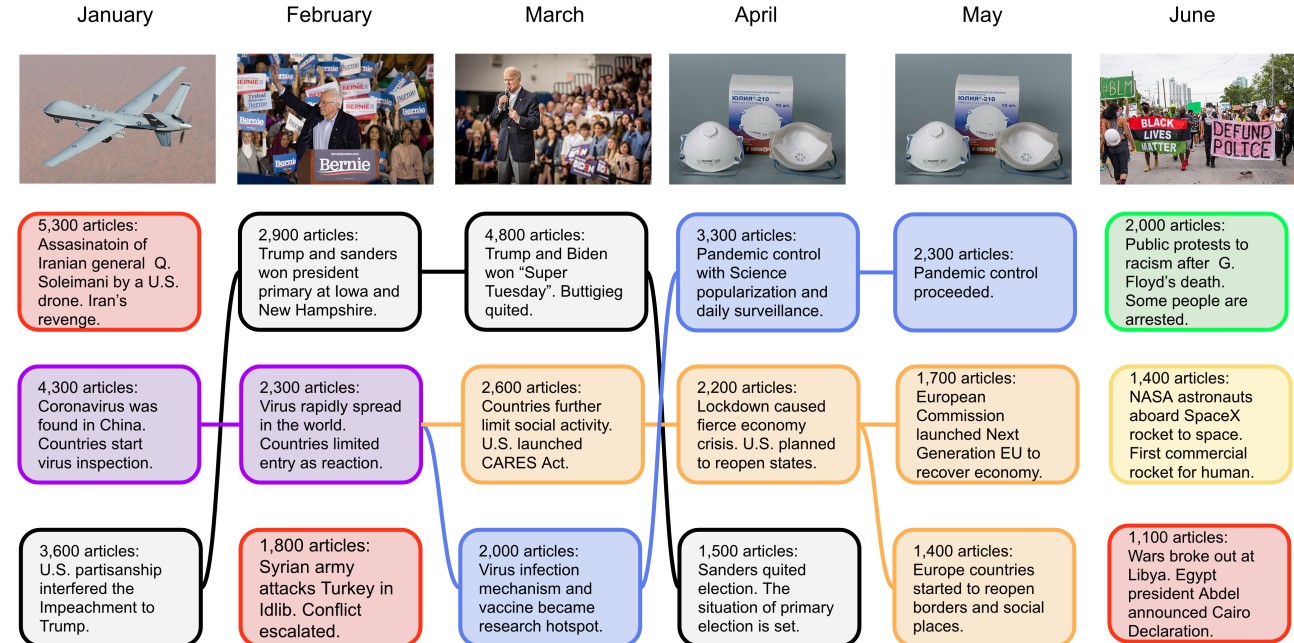

**Figure 4: The top 3 largest news event clusters in each month. The majority of these clusters are multilingual and translated into English for interpretation. Color corresponds to different branches of an event—U.S Presidential primary elections (black), military operations (red), pandemic (purple) and its two derivative branches: pandemic control (blue) and economic recovery (orange), anti-racism protests (green), and SpaceX rocket launches (yellow). Photos are from Wikipedia. Colors, connections, and descriptions are based on an interpretation of one of the authors.**

by 100 to 500 news articles. Finally, most events spanned multiple languages and nearly half of events were covered in more than 5 languages (out of 10 languages that we studied).

*4.3.3 Evaluation of news event clusters.* We evaluate the quality of the identified news events. To begin with, we find that two articles sharing a cluster are more likely to be similar than two articles sharing a named entity, which validates the effectiveness of event clusters in aggregating similar news to a degree (see details in Appendix C).

To evaluate the coherence of the detected news events, we devised an intrusion task, which is commonly used to evaluate topic models [12]. In this task, human evaluators are presented with 11 articles, all but one sampled from the same cluster. If the "intruding" article can be identified consistently, the cluster is considered to be semantically coherent. Three annotators evaluated 40 clusters. For this evaluation, we chose the 20 largest clusters and 20 random clusters. For each news cluster, evaluators were presented with the titles and URLs of the 11 corresponding articles.

We recorded an average precision of 85.8% across the annotators, which is bolstered by substantial inter-rater agreement (0.962 Gwet's AC1 and 0.809 Krippendorff's alpha coefficient). The annotator that spent the most time on the task, achieved 97.5% precision. Compared to analogous results for topic model evaluation [12], these precision values are relatively high, which lends confidence in the coherence of the news event clusters.

*4.3.4 News event interpretation.* Figure 4 shows summaries of the top 3 events with the most news articles published each month in the first half of 2020. Consecutive events are sometimes related and can form simple narratives (color-coded in the figure).

For instance, between February and March 2020 the events that attracted the most news articles were the US presidential primaries (black rectangles in the figure). During this period it was unclear who the Democratic party nominee would be. Among Democratic candidates Bernie Sanders led the primaries in February, but Joe Biden later won the key "Super Tuesday" event in March. Interest in the primaries quickly decreased in April after Sanders left the race leaving Biden as the only competitive candidate for the Democratic nomination. During that time, news around the COVID-19 pandemic evolved from the identification of the virus in China in January (violet rectangles), through realizations of its rapid transmissibility in February and March, to non-pharmaceutical interventions in April and May.

## 5 NEWS COVERAGE ACROSS COUNTRIES

Being able to identify news events at global scale enables addressing outstanding questions on how news differ across countries and flow between them. Next, we leverage country characteristics that media scholarship hypothesized as factors determining news coverage (§5.2). Namely, we use the hypothesized country characteristics as predictors in regression models of *news diversity* within a country and *news synchrony* between countries. We compare two ways of computing news diversity and synchrony: a naive baseline approach

| Type | Predictor |
|------|-----------|
| Econ. | Trade [18, 30, 37, 44, 63, 81] |
| | GDP [27, 30, 44, 62, 63] |
| | Financial Investment [44] |
| Pol. | Democracy index [38, 53, 60, 63] |
| | Political blocs |
| | Press freedom [78, 81] |
| | Mode of government [32, 77] |
| | Diplomatic ties [29, 30, 44] |
| | Military strength index [30, 38, 62, 81] |
| | Conflict intensity (global peace index) [62, 63] |
| Soc. | Population [18, 27, 37, 38, 44, 63] |
| | Immigration [35, 62, 63, 81] |
| | Population density [18] |
| | Gini index [63] |
| Geo. | Distance [18, 27, 37, 38, 63, 64] |
| | Neighbors [27, 30, 37, 62, 81] |
| | Area [27, 44, 63] |
| | Continent [18, 30, 38] |
| Cul. | Language [18, 37, 38, 63, 64, 81] |
| | Religion [44, 63, 64] |
| | Media outlets number [24, 63, 80, 81] |
| | Literacy rate [18, 44, 63] |
| | Internet user rate |

Table 2: Predictors considered in the regressions of news diversity and synchrony, including the references to studies that mention them. We have not identified references mentioning political blocs and Internet user rate.

that only uses global news similarity network (§5.3) and our main approach that also uses the global news events (§5.4), showing empirically that in comparison to the former, the latter method results in larger explanatory power of models regressing news synchrony and diversity.

## 5.1 Regression structure and representativeness

For each country and country pair, we compute a value of news diversity and synchrony. Then, we regress this value against many predictors, described in the next subsection. To mitigate potential news data biases across countries, we weight all countries equally. For our news diversity models, we ensure an equal number of news articles is sub-sampled from each of the 124 countries. Meanwhile, in our news synchrony models, we use country pairs as our samples, resulting in a sample size of 7626.

## 5.2 Factor selection and preprocessing

We surveyed extensive literature for traits of countries and international relations that are related to similar news practices, and categorized them into five types: economic, political, societal, geographic and cultural [44, 63]. All considered factors are listed (color-coded by the type) in Table 2 with detailed explanation and literature review in Appendix D. In addition to factors identified by the literature, we took into account two additional factors: the Internet user rate, to account for the relevance of online news in today's media landscape, and whether countries are members of three major blocs—NATO, EU, and BRICS—given their importance

in geopolitics. Country indices are sourced from the World Bank and The Organisation for Economic Co-operation and Development (OECD).

*5.2.1 Feature preprocessing.* Each factor is either numeric or categorical. Categorical factors are represented as dummy vectors. For news synchrony analysis, we categorize all numeric factors. Subsequently, the category combination of a country pair is treated as the pairwise category and is also represented using dummy vectors. We re-scaled all numeric predictors and the dependent variables (diversity and synchrony) so that they take values between 0 and 1.

*5.2.2 Feature and model selection.* To avoid co-linearity, we applied feature and model selection techniques. Namely, we employ two well-known criteria—Variance Inflation Factor (VIF) and Akaike information criterion (AIC)—and report results for the best models according to each criterion. We report models constructed with either of the techniques to showcase robustness of our results and methodology. VIF is a popular feature selection technique, whereas AIC is a model selection method that identifies the predictors that best explain the dependent variables. For all regressions we apply the same model and feature selection.

## 5.3 Baseline synchrony and diversity regression

A naive measure of news article synchrony or diversity could be the average news similarity between two countries of within a country, respectively. This measure, however, faces a few issues. First, it is computationally infeasible to compute the similarity between all news article pairs. We were able to compute news article similarity for only 13.6 million news article pairs, which consitutes only 1% of all pairs. If we used only these pairs for computing the average, the results would be noisy and not representative. Second, it is not clear whether such simple averages are good measures of news diversity and synchrony.

To showcase these issues and their consequences, we first perform our analysis using this baseline approach, without relying on the global news event clusters. Following this approach, to explain diversity of news, we regress the average pairwise similarity of news published in that country against the country characteristics. To explain news synchrony, we regress the average similarity of news published between a country pair. The resulting regression models achieved very low adjusted $R^2 = 0.13$ for news diversity and relatively low $R^2 = 0.30$ for news synchrony (for the AIC). By contrast, our main method based on global events (described in the next subsection) achieved adjusted $R^2 = 0.446$ or above, which is notable and significantly surpasses that of similar works in social science studies [30, 81]. In other words, the baseline model explains much less variance in the similarity and diversity measures than the model based on global events. That said, when comparing the coefficients of the baseline models to our primary regressions based on global news events, we find qualitatively similar results (see Appendix E).

## 5.4 News diversity and synchrony measures

By anchoring the news diversity and synchrony measures in global news events, we progress from an analysis rooted in 13.6 million news article pairs to an analysis encompassing 1.4 billion article

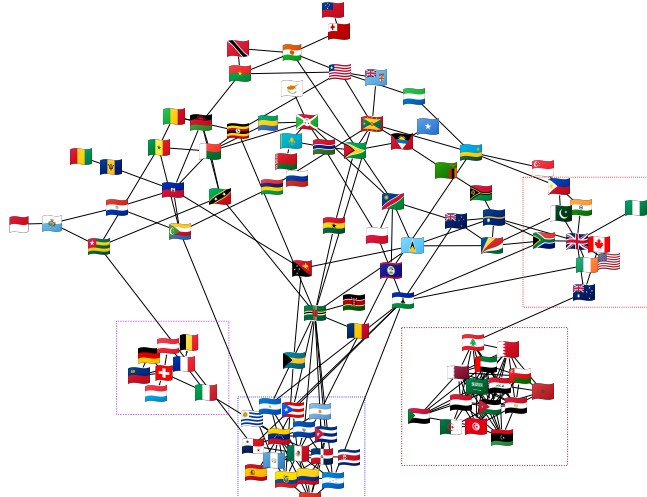

**Figure 5: News synchrony graph backbone for the top 100 countries with the largest populations. Each edge represents 95% confidence that the respective news synchrony is non-random [66]. Some countries are not selected into the graph backbone, e.g. China. The main clusters are marked with squares: the US and the UK and its former colonies (red square), 5 countries of the old European Union of 1958 (purple), Latin America and Spain (blue), the Arab world (brown).**

| Type | Predictor | VIF Cluster similarity model coeeficient | AIC Cluster similarity model coeeficient |
|---|---|---|---|
| | Intercept | 0.347 | 0.414 |
| Pol. | Federalism? | 0.118** | 0.069** |
| | Within NATO? | 0.073** | 0.044** |
| | Within EU? | −0.028** | −0.093** |
| | Within BRICS? | 0.006** | 0.012** |
| | Democracy index | | 0.175** |
| | Peace index | | −0.195** |
| Soc. | **Gini index** | **0.174**** | **0.196**** |
| | Population | 0.146* | 0.102* |
| | Net Migration | | 0.027** |
| Geo. | Area | | 0.093** |
| Cul. | #media | −0.148** | −0.130** |
| | Same media? | −0.069** | −0.063** |
| | #languages | 0.212** | 0.025** |
| | #language families | 0.058** | 0.038** |
| | Religion diversity | 0.154** | 0.126** |
| | **Internet user ratio** | **0.300**** | **0.371**** |
| | Literacy Ratio | | −0.126** |

**Table 3: News diversity regression coefficients. Positive coefficients mean more news diversity. Predictor color corresponds to its category. The top 2 predictors (according to both the VIF and AIC models) are in bold. Missing predictors or values indicate they are dropped during feature selection process. ** and * symbols stand for 0.005 and 0.01 confidence intervals respectively.**

pairs that are within clusters. To this end, for each global news event, we compute the percent of a country's news articles reporting on it. In this way, we form a distribution of articles over news events for each country. Then, we define the *news diversity* within a country as Shannon's entropy of this country's distribution [67]. Higher entropy indicates greater diversity.

To measure the similarity of the events covered by news media across countries, we introduce *news synchrony*. We define news synchrony between a pair of countries as the negative Jensen-Shannon divergence of their respective event distributions [22]. The more negative the divergence, the more synchronized the countries' media. We chose the term news synchrony over similarity because each cluster encompasses a specific temporal span, and thus captures both semantic and temporal alignment of news coverage.

## 5.5 News diversity within countries

The 5 countries with the highest internal news diversity are the U.S.A., the United Kingdom, Canada, Switzerland, and Mexico.

To explain the news diversity of each country, we regress it against country-level characteristics. The adjusted $R^2$ of the regression models are 0.494 (for the best model with factors selected through VIF) and 0.536 (for the best model selected via AIC). Table 3 lists the coefficients of the factors. The results indicate that the Internet user rate within a country is the strongest predictor of its news diversity. Internet adoption diversifies news consumption within a population, probably due to the need of news outlets to cater diverse interest and motivations of audiences [43]. The second strongest predictor is the number of official languages and language families within a country; similarly, religious diversity is a strong

predictor, measured as the Shannon entropy of the distribution over religions prevailing in a country. Overall, diverse culture is related to diverse coverage of news events.

Among societal factors, a higher Gini index indicates greater income inequality of the population and different incomes correspond to different news consumption patterns [1, 34]. Gini index is the third most predictive factor. Similarly, a larger population is related to diversity of interests in news events. Among political factors, federalist government mode is a predictor of diverse news, perhaps because the policy aims of state governments are often different from the national government [3], which poses political power to media outlets and therefore decentralizes the media focus. Unsurprisingly, news articles produced by the same media outlet are more likely to be similar.

## 5.6 News synchrony between countries

Figure 5 depicts the backbone of the international news synchrony network, extracted using an algorithm identifying non-random edges [66]. In our case, edges indicate synchrony between the countries. The graph reveals four recognizable groups: Latin America, the Arab world, five countries of the old European Union of 1958 (plus Lichtenstein, Austria, and Switzerland), and a group including the US, the UK, and its former colonies. These groups align with the geographic-linguistic communities in [38], which examined the structure of international news flow by regressing the number of international newspapers and trade amount between countries

| Type | Predictor | VIF Cluster model coefficient | AIC Cluster model coefficient |
|---|---|---|---|
| | Intercept | 0.238 | 0.247 |
| Econ. | **Trade** | **0.132**\* | **0.155**\*\* |
| | GDP (high–high) | 0.073\* | 0.073\* |
| | GDP (low–high) | 0.011\*\* | |
| Pol. | Same government mode | 0.010\* | |
| | Both federalism | 0.059\* | 0.057\*\* |
| | Federalism and other | 0.010\* | |
| | Both in NATO | 0.051\* | 0.035\* |
| | Both in BRICS | 0.095\* | 0.086\* |
| | Across NATO–BRICS groups | 0.046\* | 0.060\* |
| Soc. | Same Gini index category | 0.020\* | |
| | Gini index (high–high) | −0.037\* | −0.042\*\* |
| | Gini index (low–high) | −0.018\* | |
| | Democracy index (high–high) | 0.076\* | 0.068\*\* |
| | Democracy index (low–high) | 0.023\* | |
| Geo. | Neighbor | 0.016\* | |
| | Same Continent | 0.012\* | 0.022\*\* |
| Cul. | **Same language** | **0.125**\* | **0.139**\*\* |
| | Speak English | −0.118\* | −0.130\* |
| | Speak German | 0.014\* | |
| | Speak French | −0.079\* | −0.084\* |
| | Speak Arabic | 0.026\* | |
| | Speak Italian | −0.135\* | −0.125\* |
| | Speak Russian | −0.036\* | |

**Table 4: News synchrony regression coefficients. Positive co-efficients mean more news synchrony between a pair of countries. Predictor color corresponds to its category. The top 2 predictors (according to both the VIF and AIC models) are in bold. Missing predictors and values indicate they are dropped during feature selection process. GDP is binarized as "high" (>$500 billion) and "low" (<$500 billion). Democracy index is encoded as "high" (>5) and "low" (<5). ** and * symbols stand for 0.005 and 0.01 confidence intervals respectively.**

against country characteristics. The most-connected countries include France, Canada, Belgium, Mexico, Switzerland, and the United Kingdom. These nations consistently rank among the top 10 when evaluated using both PageRank [54] and betweenness centrality.

Next, we regress news synchrony against the characteristics of each country pair (Table 4). The adjusted $R^2$ of the models are 0.446 (VIF) and 0.448 (AIC). Trade volume is the strongest predictor, which corroborates prior findings [63, 79]. While sharing a common language is associated with higher news synchrony in general, this effect varies for specific languages.[5] On the one hand, countries speaking German or Arabic showed more news synchronization, which might be attributed to their similarities in values, lifestyles, and social norms. On the other hand, there is much less news synchronization between countries speaking English or French, likely due to the diversity of countries officially using these languages.

---

[5]The effect for a particular country can be obtained by summing the "Same language" coefficient with the coefficient of a particular language.

Speaking Italian and Russian also correlates with less news synchronization. Italian-speaking countries are relatively idiosyncratic, e.g., Switzerland is arguably not influenced much by Italy, while Vatican has a very particular role. On the other hand, Russian-speaking countries, by the virtue of recent conflicts, may have different news in their focus.

Sharing a border, or being broadly located in the same continent, intuitively correlates with news synchrony. However, controlling for these factors, geographic distance does not show a significant effect on news synchrony—a result that may be attributed to transportation, globalization and the role of the Internet [30]. The countries with the same government mode are more likely to synchronize with each other in the news event coverage, especially those with federalist systems, which may indicate that these countries share public interests, ideology, or face similar social issues.

Interestingly, countries that belong to NATO experience more news synchrony, possibly because they have common security concerns and some of the largest events correspond to military operations (Figure 4). Countries belonging to BRICS exhibit more synchronized news, possibly due to their common developmental interests. A similar effect is not evident for the EU members, which may be the result of the recent resurgence of nationalist feelings among its member states.

## 6 CONCLUSIONS

This work takes a first step towards the systematic study of news media similarity at a global scale. While correlational in nature, our work opens up avenues for future studies on the relationships between agenda setting and geographic, economic, social, political, and cultural factors. Furthermore, our computational methodology for processing global news article data enables future research to explore important historic event, such as war outbreaks, at scale across potentially long time periods. Such research could offer a more comprehensive understanding of the complex interplay between media, public opinion, and international relations.

By systematically measuring news diversity and news synchrony, and by analyzing their relationship with the country characteristics and international relations, our findings contribute to computational media scholarship. This work provides empirical evidence that substantiates our understanding of the global news ecosystem, and unpacks the factors that influence news coverage within and across countries. In particular, media scholars signaled a decline in global news diversity [36, 39]. Our results lend nuance to this phenomenon. On the one hand, the internal diversity of a country (in terms of population size, political fragmentation, and economic disparity among other factors) appears to reflect in more diverse news coverage. On the other hand, after accounting for a country's internal news diversity, nations that are structurally similar (proximate not only geographically but also culturally and socially) and maintain consistent interactions (economically but also politically) tend to synchronize their coverage of news events. Taking a broader perspective, despite the faster and increasingly globalized news cycle, the synchrony of international news reflects complex geopolitical histories and local realities. As such, this work offers methodologies to observe and interpret our continuously evolving global narrative, as seen through the lens of the news that document it.

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

## APPENDIX A: TRANSFORMER MODEL OF NEWS SIMILARITY

**News article input selection.** News articles can be quite long, yet most of the state-of-the-art deep learning models have a restricted input length, e.g., 512 word-pieces [58, 71, 85]. Therefore, we have to select a subset of the text of each news article to embed. Given that most articles follow the pyramid writing format [56, 72], the important information is often at the start; hence, we select part of the head and a part of the tail of a news. We treat decision on how much of the head vs. the tail to select as a hyper-parameter we optimize for.

**News similarity model architectures.** There are two types of mainstream encoders for text similarity: (1) cross-encoders which accept two texts as inputs and output a similarity score directly; and (2) bi-encoders which embed each input document separately as a fixed-length vector, and the similarity is then computed as the distance between two embeddings. We use a bi-encoder structure for two reasons. First, bi-encoders are more computationally efficient: the document representations for the bi-encoder may be pre-computed once and then used in all comparisons, while a cross-encoder requires computing the representations of each pair separately, thus incurring an $O(n^2)$ cost which is computational infeasible for millions of articles in our data. Second, because each document is encoded individually, bi-encoders allow for more text to included as input and thus capture more information. The 2022 SemEval task on news similarity suggested that bi-encoders could achieve comparable performance to cross-encoders [13], and this finding also mirrored our experimentation.

**News similarity mapping.** The news article similarity was labeled using a four point scale [1,4] for Very Dissimilar ($x = 1$), Somewhat Dissimilar ($x = 2$), Somewhat Similar ($x = 3$), and Very Similar ($x = 4$). We use labels of each annotator individually to train the news similarity models.

There are multiple ways to map these values into the same range as cosine similarity to compute the training loss. We can map labels from the four point scale [1,4] to range $[l, r]$ via a linear transformation $t = px + q$ without the loss of functional monotonicity. In this case two boundary conditions to news are $l = p + q$ and $r = 4p + q$ so that $p = \frac{(r-l)}{3}$ and $q = \frac{(-r+4l)}{3}$. Specifically we attempted an "unsigned" transformation $t_u$ where $l_u = 0$ and $r_u = 1$, as well as a "signed" transformation $t_s$ where $l_s = -1$ and $r_s = 1$ when we optimize our model performance.

In addition, we observed that annotators spent much more time on distinguishing Somewhat Similar or Somewhat Dissimilar pairs than identifying Very Similar and Very Dissimilar pairs, which indicates the decision boundary is vague across "Similar" and "Dissimilar." We therefore experiment with transforming the optimization objective to make its learning gradient steeper around this decision boundary, so that the model can better learn the decision boundary.

To this end, we take the cubes of unsigned and signed similarity values, $\phi = t_u^3$ and $\phi = \frac{(2t_s-1)^3}{2} + \frac{1}{2}$, respectively, where $\phi$ is the output of: (1) cross-encoders or (2) the result of cosine similarity between two news article embeddings in the case of bi-encoders.

**Multi-label learning.** In our data annotation, the news similarity is ordinal, but in realistic scenario the similarity should be a continuous, real-valued function. So annotation causes inevitably precision loss of similarity even though the annotators have a very good understanding of news similarity in their minds. We posit that the similarity ratings of other aspects (other than OVERALL) may also reveal information as to overall similarity of a news pair. By introducing them into the learning objective, precision loss from annotation can be mitigated. Specifically we devised two integrated similarities $y_1$ and $y_2$ for training, which incorporate the average of multiple aspects for a single pair:

$$y_1 = w_1 * x + (1 - w_1) * \overline{x_i}, \tag{1}$$

where $x_i$ is the similarity of the news pair in dimension $i$, $y$ is the overall similarity of the news pair, $i \in \{$ENT, NAR$\}$, and

$$y_2 = w_2 * x + (1 - w_2) * \overline{x_j} \tag{2}$$

where $j \in \{$GEO, ENT, TIME, NAR, STYLE, TONE$\}$. The weights $w_1$ and $w_2$ can be optimized by empirical study of model performance. During the training process, the learning objective of the model is to minimize the difference from its output to the integrated similarity, i.e. min $|y_i - \phi|$ where $i \in \{1, 2\}$.

**Experiments.** We optimized the selection of the proposed approaches and corresponding hyper-parameters according to model performance. In this way we identify the best model for our news article similarity task.

We take 64% of our dataset for training, 16% for development, and 20% for testing. We evaluated multiple large pre-trained multilingual deep learning models as the base of comparison; these models are the state-of-the-art for textual similarity: LaBSE [21], MPNet [71], and MUSE [85]. Also we compared two state-of-the-art models in the specific news comparison task: HFL [84] and Gatenlp[70]. Table 1 shows the summary. All models are trained on 4x NVIDIA Tesla M40, and the news similarities at global scale are inferred on 8x NVIDIA GeForce GTX 2080 Ti.

LaBSE outperformed the other base models, perhaps supported by its rich multilingual pre-training data. In general, all base models perform much worse than the fine-tuned models, which reveals the uniqueness of the news similarity task—i.e., the task is not just measuring textual similarity—and thus the need of developing a dataset for this task. By empirical test, we found the best hyperparameter configuration of our multiligual news similarity model is to take the first 456 wordpieces and last 56 wordpieces from each news article, with the basis of LaBSE model. The best performance was found using the "unsigned" range for embedding space with $t_u$ and use $y_1$ as the integrated similarity for learning with $w_1 = 0.8$. Our best bi-encoder model is able to achieve comparable performance to the HFL cross-encoder model while being an order of magnitude times faster.

## APPENDIX B: HEURISTIC REDUCING THE NUMBER OF NEWS ARTICLE PAIRS

These thresholds were selected to optimize the proportion of similar news article pairs (Figure 6), while still remaining computationally viable. The sample percentage represents the percentage of news article pairs that are labeled as (somewhat) similar in all the pairs with the same threshold (publish date interval or Jaccard similarity of named entities).

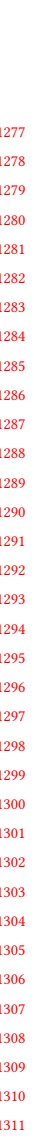
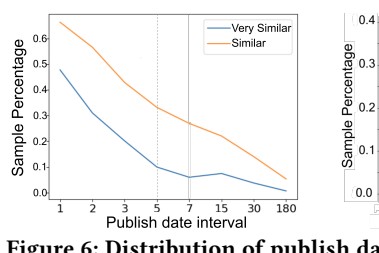
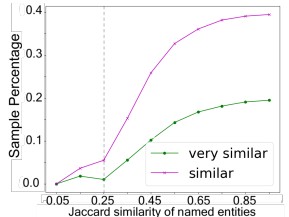

**Figure 6: Distribution of publish date intervals (left) and Jaccard similarity of named entities (right) for the news article pairs labeled as (somewhat) similar by human annotators in the extended dataset of Chen et al. [13].**

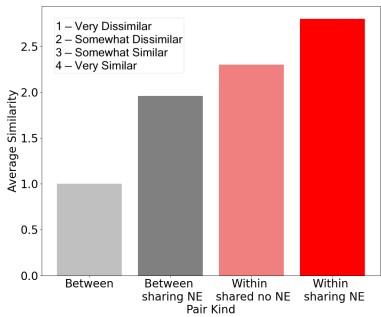

**Figure 7: Average similarity of news article pairs between and within clusters, sharing or not sharing named entities (NE), using the similarity scale of Chen et al. [13].**

## APPENDIX C: SIMILARITY OF NEWS BETWEEN AND WITHIN NEWS EVENT CLUSTERS.

As a supplementary evaluation of the quality of the news clusters, we briefly discuss their face validity and robustness. To assess face validity, we analyzed ground-truth human assessment of news similarity within and between clusters (as shared in the news similarity dataset by Chen et al. [13]).

News article pairs that share no named entities but are within the same cluster, exhibited higher average similarity than the news pairs that share named entities but belong to different clusters (Figure 7). This suggest that sharing a cluster is a more potent indicator of news article similarity than sharing a named entity.

## APPENDIX D: PREDICTORS USED TO EXPLAIN NEWS COVERAGE ACROSS COUNTRIES

We surveyed extensive literature for traits of countries and international relations that are related to similar news practices, and categorized them into five areas: economic, political, societal, geographic and cultural [44, 63]. Explanation for each area are presented in the rest of this section. Each factor is either numeric or categorical, as described in Table 5.

**Economic.** Economy has been widely found as a strong factor to influence media content and news flow in prior studies [44, 63, 79]. On the one side, financial limitations of media outlets, particularly their dependence on advertising revenue, play a major role in shaping their editorial policies [51]. A typical indicator of national economy level can be gdp [29, 62]. On the other side, modern news organizations become market-oriented and thus are often more focused on covering stories that will generate the most clicks, views, or ad revenue. For example, a drop of United States in Chinese trade volume would also lead to a decrease in its attraction in Chinese news coverage and its agenda-setting power on Chinese media. Also low income countries usually have different developing needs for international news from each other [80].

Specifically economic factors include one national trait – gdp[27, 30, 44, 62, 63]; and two international relatedness – trade [18, 30, 37, 44, 63, 81] and investment [44].

**Political.** The governments can manipulate journalistic output with their regulatory and legislative powers. Also the reporters rely on dominant social and political institutions for routine access to a significant volume of newsworthy information so that their deadlines and demands can be met [28]. Therefore how the countries distribute and execute political power internally or externally also play a role in shaping the perspectives and agendas of news coverage. Furthermore military also affects the media agenda as it guarantees the power execution procedure. Military aids to a country can support its national power while conflicts with or within a country may threaten its government control.

Specifically political factors include national traits and international relatedness in terms of political freedom (democracy index) [38, 53, 60, 63], press freedom [78, 81], nation hierarchy (republic and federalism) [32, 77], diplomatic [29, 30, 44], military power(military strength index)[30, 38, 62, 81], and conflict intensity(global peace index) [62, 63].

**Societal.** As the foundation of society, people ourselves naturally become a indispensable part in the media system. When news are spread, not only the editors and journalists decide news coverage policy [28], but also the audiences' attention form the media market and thus affect the newsworthiness [76]. Population size and immigration size are two typical indicators of people diversity, and population density can decide the frequency and strength of social connection between people. Social inequality contribute audience diversity too as individuals from different socioeconomic classes may have different interests[1, 8], which can be reflected by gini index.

Specifically societal factors include population [18, 27, 37, 38, 44, 63], immigration[35, 62, 63, 81], density [18], gini index [63].

**Geographic.** Geographic size and location of a country determines the pattern of its internal and external communications and thus shape its news coverage. When events occur far away from the media outlet's base, reporters may face logistical and financial challenges in covering them, which limit the amount and quality of coverage. Additionally, distance can create language and cultural barriers that may make it harder for journalists to fully understand and accurately report on events.

Specifically geographic factors include area [27, 44, 63], distance [18, 27, 37, 38, 63, 64], border [25, 27, 30, 37, 62, 81], continent or region [18, 30, 38]

**Cultural.** One way that culture shape media agendas is through the advocated values and norms within a particular society. For example, some cultures have a strong emphasis on community and

social responsibility, which may lead media outlets to prioritize stories about local issues and social justice. While some other cultures may have a greater emphasis on individualism and consumerism, which may result in more media attention to lifestyle and entertainment. Culture can also influence the way that news stories are framed and interpreted by media outlets, depending on factors such as language, symbolism, and historical context.

Specifically cultural factors include language [18, 37, 38, 63, 64, 81], religion [44, 63, 64], literacy rate [18, 44, 63], Internet user rate, media numbers [24, 63, 80, 81].

## APPENDIX E: BASELINE NEWS SYNCHRONY MEASURE

We mirrored the analysis in §5 using the pairwise news similarity and obtained qualitatively similar results, as shown in Table 6 and Table 7 (blank values indicate the factors are not excluded from the model by the feature selection process).

| Type | Predictor | Type |
|---|---|---|
| Econ. | Trade [18, 30, 37, 44, 63, 81] | numeric |
| | GDP [27, 30, 44, 62, 63] | numeric |
| | Financial Investment [44] | numeric |
| Pol. | Democracy index [38, 53, 60, 63] | numeric |
| | Political blocs | categorical from {NATO, EU, BRICS} |
| | Press freedom [78, 81] | numeric |
| | Mode of government [32, 77] | categorical from {Federalism, Unitary republic, Other} |
| | Diplomatic ties [29, 30, 44] | categorical from {Ambassador Relation, Other} |
| | Military strength index [30, 38, 62, 81] | numeric |
| | Conflict intensity (global peace index) [62, 63] | numeric |
| Soc. | Population [18, 27, 37, 38, 44, 63] | numeric |
| | Immigration [35, 62, 63, 81] | numeric |
| | Population density [18] | numeric |
| | Gini index [63] | numeric |
| Geo. | Distance [18, 27, 37, 38, 63, 64] | numeric |
| | Neighbors [27, 30, 37, 62, 81] | categorical from {Neighbors, Other} |
| | Area [27, 44, 63] | numeric |
| | Continent [18, 30, 38] | categorical from {Asia, Africa, North America, South America, Antarctica, Europe, and Australia} |
| Cul. | Language [18, 37, 38, 63, 64, 81] | categorical from {English, Spanish, German, French, Chinese, Polish, Turkish, Italian, Arabic, Russian} |
| | Religion [44, 63, 64] | categorical from {Christian, Muslim, Unaffil, Hindu, Buddhist, Jewish, Folk religion, Other religion} |
| | Media outlets number [24, 63, 80, 81] | numeric |
| | Literacy rate [18, 44, 63] | numeric |
| | Internet user rate | numeric |

Table 5: Predictor types considered in the within-country and between-country models with references to studies that mention them.

| Predictor | VIF Cluster similarity model coeeficient | VIF Cluster similarity model p-value | AIC Cluster similarity model coeeficient | AIC Cluster similarity model p-value |
|---|---|---|---|---|
| Intercept | 0.653 | 0.003 | 0.586 | 0.005 |
| Republic? | | | | |
| Federalism? | **-0.1178** | **0.002** | **-0.0692** | **0.002** |
| Within NATO? | -0.726 | 0.002 | -0.044 | 0.002 |
| Within EU? | 0.028 | 0.004 | 0.093 | 0.007 |
| Within BRICS? | -0.006 | 0.003 | -0.012 | 0.003 |
| GDP | | | | |
| Gini index | -0.174 | 0.003 | -0.196 | 0.003 |
| Population | -0.146 | 0.007 | -0.102 | 0.010 |
| Net Migration | | | -0.027 | 0.006 |
| Area | | | -0.093 | 0.007 |
| Democracy index | | | -0.1750 | 0.004 |
| Peace index | | | 0.1950 | 0.004 |
| #media | 0.148 | 0.004 | 0.130 | 0.004 |
| Same media? | 0.069 | 0.001 | 0.063 | 0.001 |
| #languages | -0.2118 | 0.000 | -0.0253 | 0.000 |
| #language families | -0.058 | 0.004 | -0.0378 | 0.004 |
| Same language? | | | | |
| Internet user rate | -0.300 | 0.003 | -0.371 | 0.004 |
| Literacy Rate | | | 0.126 | 0.004 |
| Religion diversity | -0.154 | 0.002 | -0.126 | 0.002 |

Table 6: Predictors of within-country similarity regression model based on Ordinary least squares. Negative coefficients mean less similarity and more diversity. Predictor color corresponds to its category. The coefficients align across Cluster model and Similarity model are bold. The strongest predictor is highlighted. Missing value indicates either this factor is dropped during factor selection process.

| Predictor | VIF Cluster model coefficient | VIF Cluster model p-value | VIF Pairwise model coefficient | VIF Pairwise model p-value | AIC Cluster model coefficient | AIC Cluster model p-value | AIC Pairwise model coefficient | AIC Pairwise model p-value |
|---|---|---|---|---|---|---|---|---|
| Intercept | 0.2378 | 0.000 | 0.6269 | 0.000 | 0.248 | 0.000 | 0.6448 | 0.000 |
| Continent sim | 0.0385 | 0.000 | -0.0303 | 0.000 | 0.0372 | 0.000 | -0.0305 | 0.000 |
| Geographic distance | | | -0.0389 | 0.000 | | | -0.0393 | 0.000 |
| Same government mode type | 0.0153 | 0.000 | 0.0040 | 0.281 | 0.0128 | 0.000 | | |
| Both republic | | | | | | | | |
| Both federalism | **0.0523** | **0.000** | **0.0100** | **0.220** | **0.0527** | **0.000** | **0.0123** | **0.110** |
| Neither republic nor federalism | | | | | | | | |
| Republic and federalism | | | | | | | | |
| Republic and other | | | | | | | | |
| Federalism and other | 0.0034 | 0.384 | 0.0074 | 0.037 | | | 0.0055 | 0.053 |
| Same political bloc | -0.0319 | 0.012 | | | -0.0287 | 0.000 | | |
| Both in NATO | 0.0046 | 0.762 | -0.0082 | 0.659 | | | -0.0235 | 0.005 |
| Both in EU | | | 0.0189 | 0.324 | | | | |
| Both in BRICS | -0.0031 | 0.930 | -0.0374 | 0.221 | | | | |
| Across NATO–EU groups | | | -0.0189 | 0.273 | | | | |
| Across NATO–BRICS groups | 0.0043 | 0.819 | -0.0090 | 0.602 | | | | |
| Across EU–BRICS groups | 0.0024 | 0.891 | 0.0096 | 0.555 | | | | |
| Same GDP category | | | -0.0008 | 0.803 | | | | |
| GDP (low–low) | | | | | | | | |
| GDP (high–high) | 0.0420 | 0.000 | 0.0077 | 0.257 | 0.0426 | 0.000 | | |
| GDP (low–high) | 0.0008 | 0.813 | | | | | | |
| Same Democracy index category | | | | | | | -0.0156 | 0.005 |
| Democracy index (low–low) | | | | | | | | |
| Democracy index (high–high) | **0.0632** | **0.000** | **0.0375** | **0.000** | **0.0557** | **0.000** | **0.0376** | **0.000** |
| Democracy index (low–high) | 0.0082 | 0.181 | 0.0158 | 0.005 | | | | |
| Same language | 0.107 | 0.000 | | | 0.1091 | 0.000 | | |
| Speak English | -0.0762 | 0.000 | 0.0294 | 0.000 | -0.0777 | 0.000 | 0.0289 | 0.000 |
| Speak German | **0.0657** | **0.008** | **0.0702** | **0.002** | **0.0646** | **0.008** | **0.0705** | **0.001** |
| Speak Spanish | | | -0.0524 | 0.000 | | | -0.0528 | 0.000 |
| Speak Polish | | | | | | | | |
| Speak French | -0.0426 | 0.001 | 0.0556 | 0.000 | -0.0436 | 0.000 | 0.0560 | 0.000 |
| Speak Chinese | -0.1188 | 0.109 | -0.0305 | 0.652 | -0.1187 | 0.108 | | |
| Speak Arabic | 0.0138 | 0.243 | -0.0387 | 0.000 | | | -0.0397 | 0.000 |
| Speak Turkish | | | | | | | | |
| Speak Italian | -0.0855 | 0.251 | 0.0110 | 0.871 | | | | |
| Speak Russian | | | | | | | | |

Table 7: Predictors of between-country similarity where country pairs are taken as samples. Negative co-efficients mean less similarity and more diversity (with country pairs). Predictor color corresponds to its category. The coefficients align across Cluster model and Similarity model are bold. The strongest predictor is highlighted. Missing value indicates this factor is dropped during factor selection process. GDP is binarized as "high" (>$500 billion) and "low" (<$500 billion). Democracy index is encoded as "high" (>5) and "low" (<5)

