# OpenReview forum: "Global News Synchrony During the Start of the COVID-19 Pandemic"
_ACM.org/TheWebConf/2024/Conference — TheWebConf24_

### Official Review · Reviewer_nbHL · 2023-11-19

**Novelty:** 2
**Technical Quality:** 2

**Review:**

This paper presents a study on “Global News Synchrony During the Start of the COVID-19 Pandemic”. As such, the contribution addresses an interesting and relevant topic of the Social Networks, Social Media, and Society track of The Web Conference 2024. In general, the paper is clearly structured and well-written (except some very minor errors). In addition, the authors mention that they will release the data and code in case of acceptance, which is a definite surplus! However, the paper has severe “technical” and “conceptual” shortcomings. In first place, the paper lacks a mathematical formulation so that the overall understanding of the presented approach remains very vague. In particular, the technical details are almost entirely unclear (e.g., “By exploiting this property, we develop a heuristic to identify a smaller set of 13.6 million news article pairs that are more likely to be similar than random pairs and compute news similarity only for this reduced set of pairs. The heuristic preserves the computational viability of this study and it is based on the observation that related articles tend to mention the same named entities, e.g., people, organizations, or locations.”). Apart from that, the actual study is somewhat “uninspiring” since the results are quite (trivially) foreseeable (“similar” languages and cultures match, so what?). The overall paper the is mostly pure engineering and a genuine scientific contribution is missing. As indicated above, the technical details are very vague and, e.g., Appendix B and Appendix C do not really help understanding what is going on *in details*. As a result, the paper lacks a take-home message (at least from a scientific point of view).

In the light of the above, I recommend to reject the paper. At the same time, it might be worth considering the submission of this paper to one of the many workshops, in particular, those addressing spatio-temporal aspects.

I acknowledge that there were no responses.

**Questions:**

- What are your heuristics *in detail*?

**Ethics Review Description:**

n.a.

**Reviewer Confidence:**

3: The reviewer is confident but not certain that the evaluation is correct

**Scope:**

3: The work is somewhat relevant to the Web and to the track, and is of narrow interest to a sub-community

---

### Official Review · Reviewer_LZ1V · 2023-11-23

**Novelty:** 6
**Technical Quality:** 6

**Review:**

The paper addresses the gap in comprehensive studies of global news coverage by introducing a novel NLP methodology and overcoming some technical challenges. For the study, the authors use a dataset of 60M news articles across 10 languages from 124 countries that spans January to June 2020. Then, the study employs a transformer model for multilingual news similarity and identifies global news events to measure synchrony and diversity in global news coverage. The results provide insights into systematic differences in media agenda-setting and international relations, with implications for future media studies.


Strengths
- S1. While comprehending the global news ecosystem is imperative, its systematic analysis faces challenges due to language limitations and the vast volume of data. The paper offers insights from the analysis of 60M news articles in 10 different languages, which is impressive.
- S2. The proposed methods demonstrate solidity and undergo rigorous evaluation. They not only facilitate broader, long-term analyses across millions of articles in diverse languages but also provide valuable data for academics and media monitoring agencies.
- S3. Their findings, such as the correlation between diverse cultures and the diverse news events coverage and the stronger news synchrony among countries sharing public interests, ideology, or similar social issues rather than mere geographical proximity, are noteworthy.
- S4. The paper exhibits exceptional clarity, being extremely well-written and easy to follow.


I cannot think of any reason to reject this paper; however, I find the below points require further clarification.
- Media Cloud data. A short paragraph about the representativeness of the Media Cloud data would be appreciated.
- Were all other languages translated into English? If so, which tool or method was employed for translation?
- How was the number of clusters (i.e., the number of news events) determined in the OSLOM algorithm?
- Who were the three annotators?

**Questions:**

- In Section 5.4., the authors mention "By anchoring the news diversity and synchrony measures in global news events, we progress from an analysis rooted in 13.6 million news article pairs to an analysis encompassing 1.4 billion article pairs that are within clusters." It is unclear how those analyses can encompass 1.4 billion article pairs.

**Reviewer Confidence:**

4: The reviewer is certain that the evaluation is correct and very familiar with the relevant literature

**Scope:**

3: The work is somewhat relevant to the Web and to the track, and is of narrow interest to a sub-community

---

### Official Review · Reviewer_tpnj · 2023-11-23

**Novelty:** 4
**Technical Quality:** 3

**Review:**

The manuscript studies the news synchrony during 2020 by analyzing news articles across the globe. The authors build a machine learning model to measure multilingual news similarity and create a method to identify events. Using the identified events, the authors quantify the diversity of news coverage within the countries and the similarities between countries. Through regression analyses, the authors identify key factors that correlate with news diversity and synchrony. I find it an interesting paper; however, some aspects of the study remain unclear to me.

According to the title, the research is about the synchrony of news coverage among different countries. And I think for most parts of the paper, the focus is on news synchrony. However, the authors also include analyses of the news diversity in the paper. It's not clear how these two concepts are closely related. In fact, I think removing the diversity component is probably helpful to keep the manuscript more streamlined and focused. But if the authors really think that part is critical, then I would suggest modifying the title, abstract, introduction, and conclusion accordingly to highlight the necessity of including it.

Another concern I have while reading the paper is that, although I think the analysis is interesting, it's not obvious to me why the findings matter other than deepening our understanding of global news coverage. The factors correlated with news synchrony between countries are not very surprising, and I have a hard time coming up with the study's implications. The authors suggest that the framework can assist studies on agenda settings, but again, it's not very obvious to me how. Please elaborate.

Some of the technical details are not clear to me either. For instance, I can't find a detailed description of the transformer model the authors adopted (or built?) to quantify multilingual news similarity. The authors suggest that their proposed method is more scalable than traditional cross-encoders, which have a complexity of N square. But if I understand it correctly, after converting the news articles into vector embeddings using the bi-encoders, one still needs to compare each pair to calculate their similarity. So, the complexity is still N square. This does not solve the scalability issue. Please elaborate. Finally, choices of the thresholds to reduce the number of news articles seem arbitrary. The authors include Figure 6 in the appendix, but I don't think it justifies the choices automatically. Please provide more details on how Figure 6 is generated and what the result means.

I also have a minor question. What does the sentence "A naive measure of news article synchrony or diversity could be the average news similarity between two countries of within a country, respectively." mean?

**Questions:**

Please see my comments above.

**Reviewer Confidence:**

3: The reviewer is confident but not certain that the evaluation is correct

**Scope:**

3: The work is somewhat relevant to the Web and to the track, and is of narrow interest to a sub-community

---

### Official Review · Reviewer_k1gB · 2023-11-26

**Novelty:** 5
**Technical Quality:** 5

**Review:**

This paper studies how news coverage synchrony and varies across countries. An efficient pipeline is proposed to estimate multilingual news similarity, making cluster analysis and explaining  synchrony mechanisms. With the pipeline, the authors identify global events in 2020 and reveal the key  factors of news synchrony. Experimental results show both the effectiveness and efficiency of the pipeline.

Strengths:
S1. The experimental results are quite convincing. Plenty of data and classic statistical methods make the results quite sound and reliable.
S2. The article is well written. It’s a pleasure to read the article.
S3. The idea of a global events cluster is quite novel. With these clusters, nice experimental results emerge naturally.

Weaknesses:
W1. The time complexity is an issue. Pipelines do not easily generalize to other efforts.
W2. Diversity within countries may have a litter problem. The articles unrelated to others are eliminated. This might influence the diversity within countries.
W3. More experiments need to be taken in regression. Other explainable algorithms, such as tree-based methods, could be taken.

**Questions:**

Q1: Is it possible to reduce the time complexity? In line 432, ‘13.6 million news article pairs across 2.2 million unique news article’. On average, no more than seven nodes are connected to one node. More nodes might be eliminated.
Q2: Does the data filtering process cause bias? The articles not related to others are eliminated before experiments. However, irrelevance means diversity to some extent. Will it introduce bias to the results?
Q3: Why don’t you use tree-based methods with regression? R^2 is about 0.5 in the experiments. Typically, boost methods can get much better results. What’s more, feature importance is also an important criterion to explain the results.

**Reviewer Confidence:**

3: The reviewer is confident but not certain that the evaluation is correct

**Scope:**

3: The work is somewhat relevant to the Web and to the track, and is of narrow interest to a sub-community

---

### Decision · Program_Chairs · 2024-01-22

**Decision:**

Accept

**Comment:**

ike all reviewers, I concur that the paper addresses an important topic - addressing the challenge of examining news coverage at a global scale. The work contributes to a novel computational pipeline spanning a dataset of 60 million news articles across 10 languages and 124 countries. I commend the authors for this humongous effort. I am in agreement with reviewer LZ1V's comment about the rigorous evaluation and the systematic approach of collecting and analyzing data. k1gB also highlighted the strengths of the experimental results and the large data collection. Reviewers did raise a few concerns
 - around the implications of this work, such as why this work matters (tpnj) or what's inspiring about the findings (nbHL),
 - raised questions around generalizability (k1gB)
 - asked for additional methodological details (LZ1V, nbHL)
 - asked for alternative methodological approaches and questioned about bias in data (k1gB)

 I appreciate the authors thoughtful and sincere attempts in responding to each of these concerns, in some cases redoing additional analyses (e.g. training gradient boosting regressor) and reframing sections of the paper.

 I am happy to recommend acceptance for this work under the assumption that the authors will take the reviewer suggestion and their responses into consideration while revising the final camera ready version of this paper.